# The Prolonged Activation of the p65 Subunit of the NF-Kappa-B Nuclear Factor Sustains the Persistent Effect of Advanced Glycation End Products on Inflammatory Sensitization in Macrophages

**DOI:** 10.3390/ijms25052713

**Published:** 2024-02-27

**Authors:** Sayonara Ivana Santos de Assis, Leonardo Szalo Amendola, Maristela Mitiko Okamoto, Guilherme da Silva Ferreira, Rodrigo Tallada Iborra, Danielle Ribeiro Santos, Monique de Fátima Mello Santana, Kelly Gomes Santana, Maria Lucia Correa-Giannella, Denise Frediani Barbeiro, Francisco Garcia Soriano, Ubiratan Fabres Machado, Marisa Passarelli

**Affiliations:** 1Laboratório de Lípides (LIM 10), Hospital das Clínicas (HCFMUSP) da Faculdade de Medicina da Universidade de São Paulo, São Paulo 01246-000, Brazil; sayonara.assis@fm.usp.br (S.I.S.d.A.); leonardo.szalo@usp.br (L.S.A.); ferreira.gui@hotmail.com (G.d.S.F.); danielleribeiro@usp.br (D.R.S.); moniquemelo4@usp.br (M.d.F.M.S.); kelly.gomes@hc.fm.usp.br (K.G.S.); 2Department of Physiology and Biophysics, Institute of Biomedical Sciences, University of São Paulo, São Paulo 05508-000, Brazil; mokamoto@icb.usp.br (M.M.O.); ubiratan@icb.usp.br (U.F.M.); 3Ciências Biológicas e da Saúde, Campos Mooca, Universidade São Judas Tadeu, São Paulo 03408-050, Brazil; rodrigo.iborra@saojudas.br; 4Laboratório de Carboidratos e Radioimunoensaio (LIM 18), Hospital das Clínicas (HCFMUSP) da Faculdade de Medicina da Universidade de São Paulo, São Paulo 01246-000, Brazil; maria.giannella@fm.usp.br; 5Laboratório de Emergências Clínicas (LIM 51), Hospital das Clínicas (HCFMUSP) da Faculdade de Medicina da Universidade de São Paulo, São Paulo 01246-000, Brazil; dfbarbeiro@usp.br (D.F.B.); gsoriano@usp.br (F.G.S.); 6Programa de Pós-Graduação em Medicina, Universidade Nove de Julho, São Paulo 01525-000, Brazil

**Keywords:** advanced glycation end products, NFKB, RAGE, toll-like receptor, lipopolysaccharide, inflammation

## Abstract

Advanced glycation end products (AGEs) prime macrophages for lipopolysaccharide (LPS)-induced inflammation. We investigated the persistence of cellular AGE-sensitization to LPS, considering the nuclear content of p50 and p65 nuclear factor kappa B (NFKB) subunits and the expression of inflammatory genes. Macrophages treated with control (C) or AGE-albumin were rested for varying intervals in medium alone before being incubated with LPS. Comparisons were made using one-way ANOVA or Student *t*-test (*n* = 6). AGE-albumin primed macrophages for increased responsiveness to LPS, resulting in elevated levels of TNF, IL-6, and IL-1beta (1.5%, 9.4%, and 5.6%, respectively), compared to C-albumin. TNF, IL-6, and IL-1 beta secretion persisted for up to 24 h even after the removal of AGE-albumin (area under the curve greater by 1.6, 16, and 5.2 times, respectively). The expressions of *Il6* and *RelA* were higher 8 h after albumin removal, and *Il6* and *Abca1* were higher 24 h after albumin removal. The nuclear content of p50 remained similar, but p65 showed a sustained increase (2.9 times) for up to 24 h in AGE-albumin-treated cells. The prolonged activation of the p65 subunit of NFKB contributes to the persistent effect of AGEs on macrophage inflammatory priming, which could be targeted for therapies to prevent complications based on the AGE–RAGE–NFKB axis.

## 1. Introduction

Advanced glycation end products (AGEs) belong to a family of structurally heterogeneous compounds formed by the Maillard reaction, characterized by the covalent reaction between glucose or reducing sugars with the amino-terminal portion of lysine and arginine residues in proteins, nucleic acids, and phospholipids. The concentration of AGE in body fluids and tissues is elevated in the presence of hyperglycemia, although inflammatory reactions, oxidative stress, and impaired renal function generate oxoaldehydes that react with macromolecules, leading to a very rapid formation of AGEs, independently of diabetes mellitus (DM) [1]. Additionally, exogenous AGEs can be obtained from dietary sources based on food composition and cooking methods, tobacco, and pollution [1,2].

Advanced glycation end products are linked to the pathophysiology of various non-communicable chronic diseases through signaling via the receptor for AGE (RAGE), which promotes the generation of reactive oxygen species (ROS) by activating nicotinamide adenine dinucleotide phosphate oxidase (NADPH oxidase) and the mitochondrial respiratory chain. The subsequent activation of the NF-kappa-B nuclear factor (NFKB) guides the transactivation of inflammatory genes and perpetuates a vicious circle by stimulating the expression of *Ager* that encodes RAGE [3,4]. Moreover, other transcription factors, such as activator protein 1 and forkhead box protein O4 via various signal transduction cascades, such as mitogen-activated protein kinases (MAPK), c-Jun N-terminal kinases, extracellular signal-regulated protein kinases 1 and 2, and Janus kinase/signal transducers and activators of transcription (JAK-STAT), are elicited by the AGE–RAGE axis [5].

A converging signaling pathway between RAGE and toll-like receptor 4 (TLR4) is evident, amplifying cellular inflammatory sensitization by AGE [6]. Advanced glycated end products sensitize macrophages to inflammatory stimulation by lipopolysaccharide (LPS), a natural ligand for TLR4, promoting the expressions of inflammatory cytokines and chemokines [7,8]. In this context, AGEs seem to act as modulators of the inflammatory process that guides atherogenesis, insulin resistance, and diabetes mellitus, among others.

Inflammation is aggravated by disturbances in cellular cholesterol homeostasis induced by AGEs, especially by the reduction in the ATP-binding cassette transporter A1 (ABCA1) content [8,9]. The accumulation of cholesterol and toxic oxysterols conveys inflammatory pathways related to atherogenesis and plaque instability. AGE plasma levels are surrogate markers for advanced and unstable human atherosclerotic lesions [10].

Advanced glycated albumin (AGE-albumin) induces lipid accumulation and enhances the expressions of carboxymethyl lysine, RAGE, and 4-hydroxynonenal, a marker of lipid peroxidation in the aortic arch of non-diabetic dyslipidemic mice [11]. In healthy rats, chronic intraperitoneal administration of AGE-albumin favored adipocyte hyperplasia and the infiltration of inflammatory macrophages in peri-epididymal adipose tissue, both associated with a reduction in the expression of glucose transporter 4 (Glut4) [12]. Moreover, in the same animal model, an increased inflammatory profile was observed in skeletal muscle with the induction of endoplasmic reticulum stress together with an increased expression of nuclear NFKB, repressing the expression of the *Scla2a4* gene that encodes Glut4 [13]. These findings contribute to the diminished systemic insulin sensitivity in those animals, pointing to the role of AGEs in the development of insulin resistance and type 2 DM, based on the development of an inflammation milieu.

The NFKB consists of a small family of transcription factors that regulate various cellular functions, especially those related to adaptive and innate immune function. This family comprises five structurally related proteins: NFKB1 or p50, NFKB2 or p52, RelA or p65, RelB, and c-Rel [14]. In the non-activated basal state, NFKB is associated with the family of inhibitory cytoplasmic proteins, the NFKB inhibitor, IKB. The canonical activation pathway of NFKB, known as the NFKB essential modulator (NEMO)-dependent pathway, involves signaling through different cytokine receptors, mitogens, growth factors, and the AGE–RAGE axis. In this pathway, IKB phosphorylation occurs by a complex of protein kinases IKK [NFKB inhibitor kinase alpha (IKKA) and NFKB inhibitor kinase beta (IKKB)], particularly the IKKB subunit, which, when multimerized by NEMO, facilitates transient signals that allow for the subsequent proteasomal degradation of IKB through polyubiquitination. This process results in the release of NFKB and rapid nuclear translocation, particularly of the heterodimerizing subunits Nfkb1/RelA (p50/p65) and Nfkb1/c-Rel, which bind to gene promoter regions. This process is crucial for the expression of inflammatory genes and the production of cytokines and chemokines, with its exacerbated activation linked to various pathophysiological processes [15,16,17]. The activation of the NFKB pathway can occur directly through the AGE–RAGE axis and through a myeloid differentiation primary response 88 (MyD88)-dependent pathway, which can be activated by both the AGE–RAGE axis and the TLR4 receptor. The TLR4 receptor can activate this pathway by binding endotoxins, such as LPS. The convergence of these pathways ultimately increases the nuclear translocation of NFKB, further amplifying the inflammatory response [6,18].

The modification of cellular macromolecules by AGE is identified as one of the mechanisms responsible for metabolic memory or the legacy effect, as described in large epidemiological studies, such as the Diabetes Control and Complications Trial/Epidemiology of Diabetes Interventions and Complications (DCCT/EDIC) and the United Kingdom Prospective Diabetes Study (UKPDS) [19,20]. Alongside oxidative stress and epigenetic changes, AGEs appear to contribute to the incidence of long-term complications in previously decompensated DM, despite subsequent compensation [19,20]. In cellular models, it has been possible to characterize the temporal effect of AGEs on inflammatory sensitization in macrophages as a sustained effect that persists, even following the removal of AGEs from the culture medium. This study hypothesized that the sustained activation of NFKB subunits p50 and p65, reflected by their prolonged presence in the nuclear compartment and changes in the expression of inflammatory genes, may persist in cells stimulated with albumin-AGE and be responsible for the inflammatory sensitization of macrophages to LPS. Then, in RAW 264.7 macrophages, the durations of the action of AGE-albumin on the (1) nuclear content of NFKB subunits, p50 and p65; (2) secretions of inflammatory cytokines (TNF, IL-6, and IL-1 beta); and (3) expressions of genes involved in the AGE signaling and inflammatory response (*Nfkb1*, *RelA*, *Ager*, *Tlr4*, *IL6*, *Tnf*, *Abca1*, *Abcg1*, and *Jak2*) were addressed.

## 2. Results

The release of lactate dehydrogenase (LDH) was similar between cells treated with C- or AGE-albumin. Although decreases in LDH activity were observed at 12 h and 24 h, the values remained similar between the two experimental conditions (Figure 1).

Figure 2 illustrates the role of AGE-albumin in immediate cell sensitization (time, 0 h) to LPS, showing increased secretions of TNF (Figure 2A), IL-6 (Figure 2C), and IL-1beta (Figure 2E). Additionally, a persistent effect was observed, sustained for up to 24 h of TNF and IL-6 secretions and 12 h for IL-1beta. There were 1.56-fold, 16-fold, and 5.2-fold increases in the area under the curve (AUC), respectively, for the secretions of TNF, IL-6, and IL-1beta in cells incubated with AGE-albumin, compared to C-albumin (Figure 2B,D,F).

In incubations with the addition of HDL along with treatment with C- and AGE-albumin, it was observed that there was the same sustained cytokine release profile. Thus, under these conditions, HDL did not counteract the effect of AGE-albumin on the prolonged sensitization of macrophages to inflammation induced by LPS up to 24 h (Figure 3A,C). There were 4-fold and 3.6-fold increases, respectively, in the AUC for IL-6 and IL-1beta secretions in cells incubated with AGE-albumin, as compared to C-albumin (Figure 3B,D).

The expression of genes related to inflammation was determined at 8 h and 24 h after the removal of C- or AGE-albumin, followed by incubation with LPS. In Figure 4A, it is observed that the expression of *Il6* remained elevated at both 8 h and 24 h after the removal of AGE-albumin, following the challenge with LPS. However, the expression of *Tnf* was not altered (Figure 4B). The expression of *Nfkb1* showed no difference at 8 h and 24 h (Figure 4C); on the other hand, *RelA* showed increased expression at 8 h and normalization at 24 h (Figure 4D).

The expression of genes that encode for RAGE (*Ager*) and TLR4 (*Tlr4*) did not exhibit changes at the analyzed time points (Figure 5A,B). Since AGEs have been associated with impaired cholesterol efflux, the expressions of genes encoding the ABCA-1 and ABCG-1 receptors (Figure 5C,D), which are involved in reverse cholesterol transport, and the gene encoding the JAK-2 protein (Figure 5E), which modulates the binding of apoA-1 to ABCA-1 and inflammation, were evaluated. An increase in the expression of the *Abca1* gene (Figure 5C) was observed after a 24 h resting period in the condition previously treated with AGE-albumin, with no significant difference between treatments during the various resting times in the expression of the *Abcg1* (Figure 5D) and *Jak-2* (Figure 5E) genes.

The nuclear contents of p50 and p65 subunits of NFKB were determined by Western blot at 0 h, 8 h, and 24 h after the removal of C- or AGE-albumin, followed by incubation with LPS. The nuclear content of p50 was similar when comparing cells treated with C- or AGE-albumin during the mentioned periods (Figure 6A) and also between the time points considering the same treatment (Figure 6B,C). The nuclear content of p65 was 2.9 times higher in cells treated with AGE-albumin, compared to those exposed to C-albumin, remaining elevated up to 24 h (Figure 6D). Regarding the time points, considering the same treatment, cells treated with C-albumin showed a decrease in p65 protein only from 24 h onwards (Figure 6E). Cells treated with AGE-albumin demonstrated a decreasing profile over time, with lower concentration at 24 h (Figure 6F).

## 3. Discussion

Inflammation is an underlying mechanism in the genesis and evolution of chronic diseases, including DM, atherosclerosis, cancer, and others. Advanced glycation end products prime cells to an inflammatory response by disturbing lipid metabolism and inducing oxidative and endoplasmic reticulum stress. Moreover, by interacting with RAGE and TLR4, AGEs overlay inflammatory signaling pathways that culminate in the activation of canonical NFKB-dependent pathways. In this investigation, it was demonstrated that AGE-albumin has a persistent effect on LPS-induced inflammation in macrophages that relies on a sustained nuclear activation of the p65 subunit of the NFKB, even after cell resting in the absence of AGE.

As compared to C-albumin, AGE-albumin promptly increased the LPS-elicited secretions of IL-6, TNF, and IL-1beta, with a long-lasting effect even after the removal of AGE-albumin from the cell medium. Interestingly, similar results were previously observed regarding IL6 and TNF secretions when bone marrow-derived macrophages were incubated with albumin isolated from poorly controlled DM subjects, reinforcing that the glycation reaction that takes place in vivo and in vitro has similar biological actions [8]. Remarkably, albumin is the major serum protein undergoing modification through glycation, largely due to the abundance of lysine residues in its structure [21]. The generation of oxoaldehydes during glycemic excursions, hyperglycemia, and inflammatory conditions favors its rapid modification by AGEs. While the detrimental impact of albumin-AGE on glycemic and lipid homeostasis has been well-established, there remains a limited understanding of its role in cellular metabolic memory. The activation of the AGE–RAGE axis is necessary for lipid homeostasis impairment, and models with gene silencing of *Ager* or knockout for *Ager* show resistance to the effects of AGE-albumin [22].

In the nucleus pulposus cells of vertebral discs, it has been demonstrated that AGEs activate the pyrin domain-containing protein 3 (NLRP3) inflammasome, leading to the time-dependent productions of pro-IL-1beta, cleaved caspase-1, and IL-1beta. The longer the exposure time to AGEs, the greater the activation of the inflammasome [23]. This same activation was also observed in corneal epithelial cells of diabetic mice, where the increased accumulation of AGE in the corneal epithelial tissue was associated with higher expressions of NLRP3, caspase-1, ASC, and gasdermin, compared to the control group. The findings of this study reiterated that AGEs are a significant stimulus in maintaining the inflammatory cellular response through NLRP3 [23,24].

In the current investigation, IL-1beta remained elevated at all time points, which may be related to sustained inflammasome activation, thus explaining the effect of persistent inflammation on metabolic memory. This is because this cytokine increases the expression of cell adhesion molecules, such as vascular cellular adhesion molecule-1 (VCAM-1), intercellular adhesion molecule-1 (ICAM-1), and monocyte chemoattractant protein-1 (MCP-1), in endothelial vessel cells, which are responsible for recruiting macrophages [25,26,27].

When incubating RAW 294.7 macrophages with glycated bovine albumin at a concentration of 300 µg/mL for 15, 30, and 60 min, Wu et al. observed an increase in the cytosolic and nuclear contents of the p65 protein. The 15 min time point resulted in a pronounced translocation of the p65 subunit from the cytosol to the nucleus, reaching its maximum at 30 min and declining at 60 min [28]. Similarly, Lan et al. demonstrated that AGEs induce apoptosis in MS1 pancreatic islet endothelial cells and the dose-dependent phosphorylation of the p65 subunit in these cells. Increasing concentrations of AGE (25–200 µg/mL) for 24 h increased apoptotic cell death and the protein contents of cleaved caspase-3 and polyADP ribose polymerase. The total nuclear content of p65 was not altered among different concentrations, but the total cellular lysate content of phosphorylated p65 increased starting from 100 µg/mL [29].

In the present investigation, the nuclear content of p65 was sustained, even after 24 h of removing AGE-albumin from the culture medium, compared to the C-albumin. This may be justified by the increased secretion of cytokines (TNF, IL-6, and IL-1beta) through the initial production of NFKB, which may have sustained its production through positive feedback, as the secretion of these cytokines also contributes to the increase in intracellular NFKB. Furthermore, LPS also participates in the activation of various inflammatory pathways, contributing to cytokine production and the increase in receptors such as RAGE and TLR4, collaborating with cell sensitization to glycated albumin.

The NF-kappa-B nuclear factor, particularly its p65 subunit, can be activated by different stimuli, leading to nuclear translocation, but its pro-inflammatory role will depend on post-translational modifications or histones regulating its target genes, thus determining the strength and duration of the transcriptional response of this factor [30]. One of the important post-translational modifications is the phosphorylation of the p65 subunit in serine residues by the activities of different kinases, with major stimulators being IL-1, LPS, TNF, p53, and TAX (human T-lymphotropic virus-1-encoded protein) [31,32,33].

A higher expression of *RelA* was observed at the 8 h time point, which is associated with the increased secretion of cytokines that remained elevated at later time points in macrophages treated with AGE-albumin. When evaluating the gene expression profile encoding for NFKB at 0 h, 8 h, and 24 h, it is important to consider that, in addition to the potential modulation of this factor in a time-course manner, other variables, including post-transcriptional changes, may complicate the analysis. Regarding the genes *Abca1*, *Abcg1*, *Tlr4*, *Rage*, and *Jak-2*, which were also assessed in this study, a more detailed analysis of the protein content is also necessary to understand the transcription dynamics and the formation of receptors, cytokines, and proteins involved in the activation of the inflammatory process.

The sustained activation of NFKB and the inflammatory response is suggested as one of the underlying mechanisms for metabolic memory in rats, where normoglycemia was achieved after a prolonged period of hyperglycemia [34,35,36,37]. In vitro studies have demonstrated the persistence of expression, activity of markers of oxidative and inflammatory stress, and mediators of cell death in endothelial and retinal cells treated for 14 days with a high glucose concentration, followed by exposure to low glucose medium for 7 days [38,39]. In the retinas of diabetic rats, inadequate glycemic control induced increases in caspase-3, NFKB activity, and oxidative stress, compared to the retinas of healthy rats. In animals with DM, the reinstitution of good glycemic control after 2 months of poor control reduced these parameters, which was not observed after 6 months of prior decompensation [36,37,38,39,40].

A study by Yao and colleagues using human umbilical cord epithelial cells and mouse cardiac microvascular cells also revealed the damage caused by transient hyperglycemia in cardiovascular complications. Transient hyperglycemia, followed by glycemic normalization, led to an increase in NFKB signaling in these cells with the phosphorylation of its p65 subunit, triggering an inflammatory loop with the increased expression of microRNA 27a-3p, which negatively regulates the erythroid-2 nuclear factor (NRF-2). Consequently, it increases ROS generation, leading, again, to increased NFKB. Additionally, this study showed that the increase in ROS caused by this cellular memory is also related to mitochondrial dysfunction, increasing TGF-beta signaling and inducing epithelial–mesenchymal transition, thereby causing cardiovascular damage [41].

The increase in ROS generation in epithelial cells was also found to be responsible for the increased histone methylation (H3K4me1) in the proximal promoter region of the NFKB gene (p65 subunit), favoring the expression of inflammatory genes such as MCP1 and VCAM1. These changes persist for 6 days after the normalization of glucose concentration in the culture medium and for months in diabetic animals after the recovery of pancreatic beta cell function. Reduction of mitochondrial ROS generation and methylglyoxal-induced superoxide production prevented these epigenetic changes and increased NFKB transactivation. In animals kept in short-term hyperglycemia, an increase in histone methylation (H3K4me1) and the expression of p65 in vascular endothelial cells were still observed after adequate metabolic control [42].

Epidemiological studies in the DM population demonstrated that both micro- and macrovascular complications of DM can develop, even after inadequate glycemic control is normalized. These findings suggest the existence of a cellular memory for hyperglycemic conditions, to which AGEs contribute by altering the structures of various macromolecules, perpetuating oxidative stress and inducing epigenetic changes [43]. Indeed, the tissue burden of different AGE structures is a predictor of micro- and macroangiopathy [44], and the plasma concentration of AGE predicts the severity of coronary atherosclerotic lesions [10,45].

The findings of the current investigation have demonstrated that (1) AGEs sensitize macrophages to inflammatory stimulation promoted by LPS, leading to the increased secretion of inflammatory cytokines; (2) the inflammatory sensitization induced by AGE-albumin is prolonged, persisting up to 24 h after the removal of AGE from the culture medium; (3) the secretions of IL-6, TNF, and IL-1 beta induced by LPS after treatment with AGE-albumin indicate the activation of different inflammatory pathways, including the inflammasome system; and (4) the prolonged activation of NFKB, evidenced by increased *RelA* expression and higher nuclear content of the p65 subunit after 8 h and 24 h, respectively, of cell resting in the absence of AGE-albumin, supports the metabolic memory induced in macrophages by advanced glycation end products. Pharmacological therapies focused on reducing AGE signaling, as well as NFKB inflammatory signaling, may help to prevent the deleterious effects of AGE, mediating inflammatory stress.

## 4. Material and Methods

### 4.1. Isolation of Lipoproteins

After a 12 h fasting period, venous blood was drawn from healthy human donors (*n* = 6), and plasma was promptly obtained through centrifugation at 3000 rpm and 4 °C. All participants signed an informed consent, which was previously approved by the Ethics Committee of Hospital das Clínicas (HCFMUSP) da Faculdade de Medicina da Universidade de São Paulo (CAPPesq 2.397.639), in accordance with the Declaration of Helsinki. Preservatives were added to the plasma pool (µL/mL of plasma): 20 µL of chloramphenicol/gentamicin (0.25%) (Merck, Darmstadt, Germany), 5 µL of benzamidine (2 mM) (Sigma-Aldrich, Steinheim, Germany), 5 µL of aprotinin (0.5%; Sigma-Aldrich), and 0.5 µL of phenylmethylsulfonyl fluoride (PMSF; Sigma-Aldrich), followed by density adjustment with potassium bromide. Low-density lipoproteins (LDL; d = 1.019–1.063 g/mL) and high-density lipoproteins (HDL; D = 1.063–1.21 g/mL) were isolated by ultracentrifugation in a discontinuous density gradient [46], followed by sterilization through a 0.22 µm filter. Protein concentration was determined using the Lowry’s method [47].

### 4.2. LDL Acetylation

LDL was acetylated following the protocol described by Basu et al. [48] following extensive dialysis against the phosphate buffer (PBS) with ethylenediaminetetraacetic acid (EDTA; pH = 7.4) for 24 h at 4 °C, sterilization, and protein concentration determination.

### 4.3. Advanced Glycation of Albumin In Vitro

Bovine fatty acid-free albumin (Sigma-Aldrich) (40 mg) was incubated with 10 mM glycolaldehyde (Sigma-Aldrich) dissolved in PBS with EDTA (pH = 7.4). Control albumin (C) was prepared in the presence of PBS only. Incubations were carried out under sterile conditions in a nitrogen atmosphere and in a water bath at 37 °C, with agitation for 4 days. Subsequently, the samples were dialyzed against PBS and sterilized through a 0.22 µm filter, and protein concentration was determined. Endotoxin levels in the albumin samples were determined using the Limulus amebocyte lysate (LAL) assay (Cape Cod, Falmouth, MA, USA), and only samples with endotoxin concentrations below 50 pg/mL were used in the experiments.

### 4.4. Culture of RAW 264.7 Macrophages

Cell culture was conducted in 75 cm^2^ flasks using RPMI medium containing 1.5 g/L sodium bicarbonate, 2.5 g of glucose, 2.6 g/L Hepes, and 0.11 g/L sodium pyruvate and supplemented with 10% fetal bovine serum (Thermo Fisher Scientific, Waltham, MA, USA, Thermo Scientific—Wilmington DE, Waltham, MA, USA). For experiments, cells were detached using a scraper, seeded in 24-well plates (8 × 10^5^ cells/well) or 6-well plates (2 × 10^6^ cells/well), and incubated for 24 h for adherence in RPMI medium. Following the incubation with C- or AGE-albumin, the cell culture medium was isolated to assess the release of lactate dehydrogenase (LDH), a marker of cellular cytotoxicity (CyQUANT™ LDH Cytotoxicity Assay, Invitrogen, Waltham, MA, USA).

### 4.5. Determining the Persistence Time of the Effect of AGE-Albumin Effect on Macrophages via NFKB Activation

To determine whether the persistence of the effect of C- or AGE-albumin in inducing inflammation is linked to the activation of NFKB-dependent pathways, RAW cells were enriched in cholesterol by incubation with acetylated LDL (50 μg/mL) for 24 h, following treatment with C- or AGE-albumin (2 mg/mL) for 48 h. In some sets of experiments, HDL (50 µg/mL) was added to the incubations. After careful washing with PBS containing fatty acid-free albumin (FAFA), cells were maintained in RPMI/FAFA for 0 h, 4 h, 8 h, 12 h, or 24 h, washed again with PBS/FAFA, and challenged by incubation for an additional 24 h with LPS (*E. coli* serotype, 1 μg/mL in medium; Sigma). The culture medium was isolated and frozen at −80 °C for subsequent analysis of the inflammatory cytokines, TNF, IL-1beta, and IL-6 (ELISA (R&D System—Duo Set, Minneapolis, MN, USA). The cells were washed and used for gene expression analysis and determination of protein levels.

### 4.6. Gene Expression Analysis of Nfkb1, RelA, Il6, Tnf, Abca1, Tlr4, and Ager

The total mRNA was extracted from cells after treatment with the Trizol^®^ method (Invitrogen, Life Technologies, Carlsbad, CA, USA), following the manufacturer’s instructions. The sample concentrations were quantified using the Nanodrop instrument (Thermo Fisher Scientific), and only samples with an OD260/OD240 ratio ≥ 1.8 were utilized. mRNA expression was measured by real-time polymerase chain reaction (RT-qPCR) using fluorophore-labeled probes FAM acquired in TaqMan gene expression assays format (Applied Biosystems Inc., Foster City, CA, USA) for the following genes: *Abca1*, *Abcg1*, *Jak2*, *Tnf*, *Il6*, *Ager*, *Tlr4*, *RelA*, and *Nfkb1*. Relative gene expression was determined using the *Hprt* gene, and RT-qPCR reactions were performed on the StepOnePlus Real-Time PCR System (Applied Biosystems Inc., Foster City, CA, USA). Relative gene expression quantification was calculated using the comparative cycle threshold method (Ct; 2^−ΔΔCt^) [49].

### 4.7. Determination of Protein Content by Western Blot

Proteins were extracted using the method adapted from Andrews and Faller [50], and total protein concentration in the nuclear extract was determined by the Bradford method (Biorad Laboratories, Hercules, CA, USA). Nuclear proteins (10 µg) were separated by sodium dodecyl sulfate gel electrophoresis (SDS-PAGE), according to the method developed by Laemmli and modified by Garfin and colleagues [51]. After electrophoresis in polyacrylamide gel, proteins were electrotransferred to a nitrocellulose membrane (Biorad Laboratories, Hercules, CA, USA) overnight at 25 mA, followed by blocking with Tris-buffered saline Tween (TBST) containing 5% skim milk and 0.1% Tween for 2 h. The membranes were washed with TBST with 0.1% Tween (3 washes of 10 min) to remove excess milk. Subsequently, the membranes were incubated with primary antibodies diluted in TBST with 0.005% Tween and 3% skim milk (at dilutions recommended by the manufacturers): NFKB1 (1:1000) (Abcam, Cambridge, UK) and RELA (1:1000) (Abcam, Cambridge, UK) for 19 h at 4 °C with agitation on a rocking platform (Platform Varimix, Barnstead Thermolyne, Thermo Scientific, Waltham, MA, USA). After washing with TBST with 0.1% Tween (3 washes of 10 min each), the membranes were incubated with peroxidase-conjugated antibody (Amersham ECL Rabbit IgG, HRP-linked whole Ab from donkey (Cytiva, Marlborough, MA, USA) (1:2000)) for 1 h at 25 °C. Band visualization was performed using the enhanced chemiluminescence (ECL) method (SuperSignal West Pico Chemiluminescent Substrate; Thermo Scientific, Rockford, IL, USA), and image capture was performed with a G:BOX gel documentation system (Syngene, Frederick, MD, USA). Band intensities were quantified by optical densitometry ImageQuant TL (Amersham Biosciences UK Limited, Slough, Buckinghamshire, UK) and normalized to the optical densitometry of their respective loading controls (Ponceau S staining of the corresponding bands) [52,53]. The final results were normalized, considering the mean values of the controls as 1.

### 4.8. Statistical Analysis

Statistical analysis was conducted using the GraphPad Prism software (version 5.04) for Windows and Microsoft^®^ Excel for Mac (version 16.52). The normality of the samples was assessed using the Shapiro–Wilk test, and group comparisons were performed using unpaired Student’s *t*-test, Mann–Whitney test, or one-way ANOVA. A value of *p* < 0.05 was considered statistically significant.

## Figures and Tables

**Figure 1 ijms-25-02713-f001:**
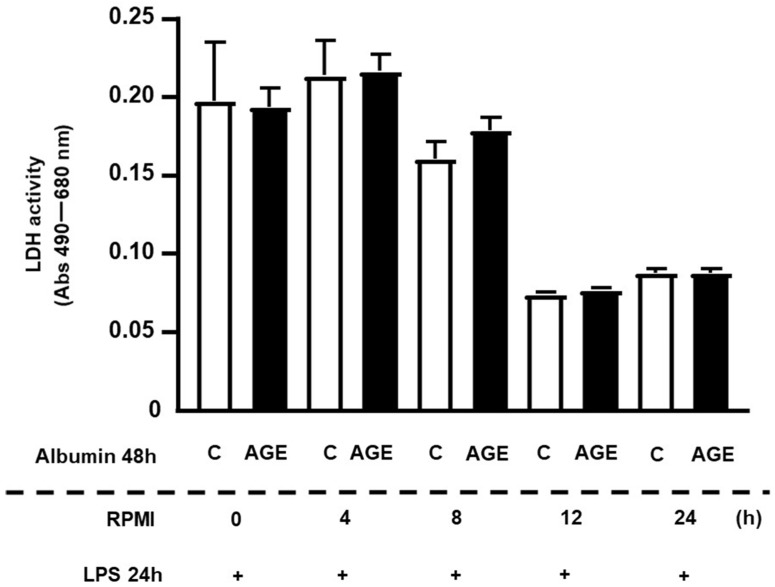
Lactate dehydrogenase (LDH) activity in RAW 264.7 macrophages treated with C- or AGE-albumin and challenged with LPS after resting for different intervals of time. RAW 264.7 macrophages were treated for 24 h with acetylated LDL and then with C- or AGE-albumin (2 mg/mL) for 48 h. After different resting intervals in RPMI/FAFA, cells were challenged with LPS for 24 h. Following the completion of treatments, the culture medium was collected for the LDH assay. Values were compared using Student’s *t*-test and represented as mean ± standard error of the mean (*n* = 6).

**Figure 2 ijms-25-02713-f002:**
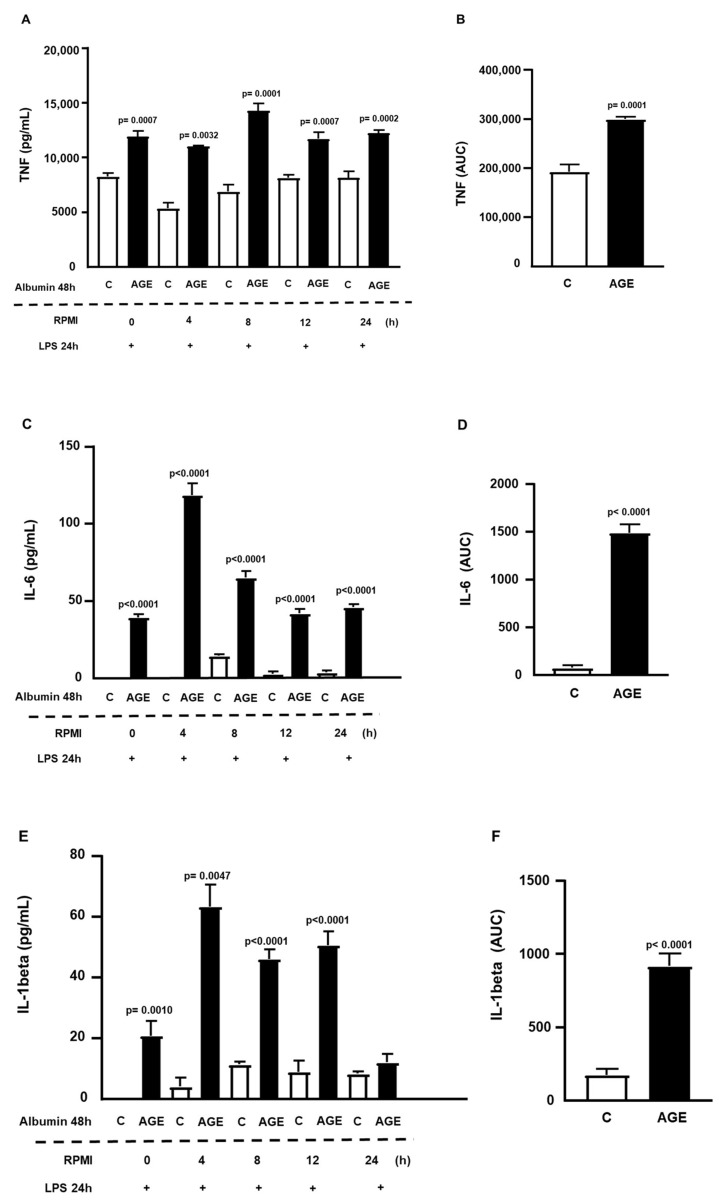
Persistence of the effect of AGE-albumin on the inflammatory response in RAW 264.7 macrophages. RAW 264.7 macrophages were treated for 24 h with acetylated LDL (50 µg/mL) and then with either C- or AGE-albumin (2 mg/mL) for 48 h. Following various resting intervals in RPMI/FAFA, cells were exposed to LPS (1 µg/mL) for 24 h. The concentrations of TNF, IL-6, and IL-1beta were determined by ELISA, with total concentrations presented in panels (**A**,**C**,**E**) and the area under the curve (AUC) depicted in panels (**B**,**D**,**F**). Normality was tested using the Shapiro–Wilk test, and comparisons were conducted using Student’s *t*-test. Results are expressed as mean ± standard error of the mean (*n* = 6 for each incubation time).

**Figure 3 ijms-25-02713-f003:**
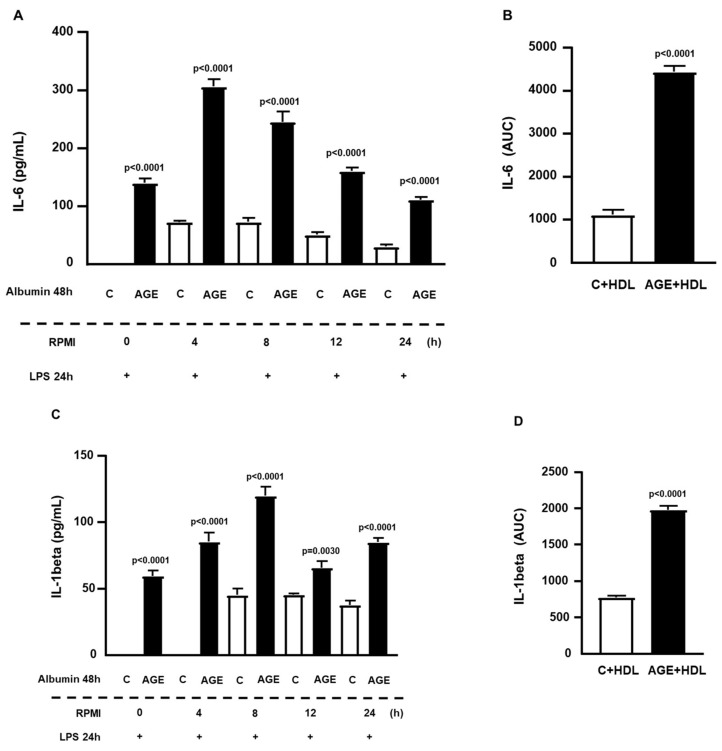
Persistence of the effect of AGE-albumin on the inflammatory response in RAW 264.7 macrophages. RAW 264.7 macrophages were treated for 24 h with acetylated LDL (50 µg/mL) and then with C- or AGE-albumin (2 mg/mL), alone or in the presence of HDL (50 µg/mL), for 48 h. Following various resting intervals in RPMI/FAFA, cells were exposed to LPS (1 µg/mL) for 24 h. The concentrations of TNF, IL-6, and IL-1beta were determined by ELISA, with total concentrations presented in panels (**A**,**C**) and the area under the curve (AUC) depicted in panels (**B**,**D**). Normality was tested using the Shapiro–Wilk test, and comparisons were conducted using Student’s *t*-test. Results are expressed as mean ± standard error of the mean (*n* = 6 for each incubation time).

**Figure 4 ijms-25-02713-f004:**
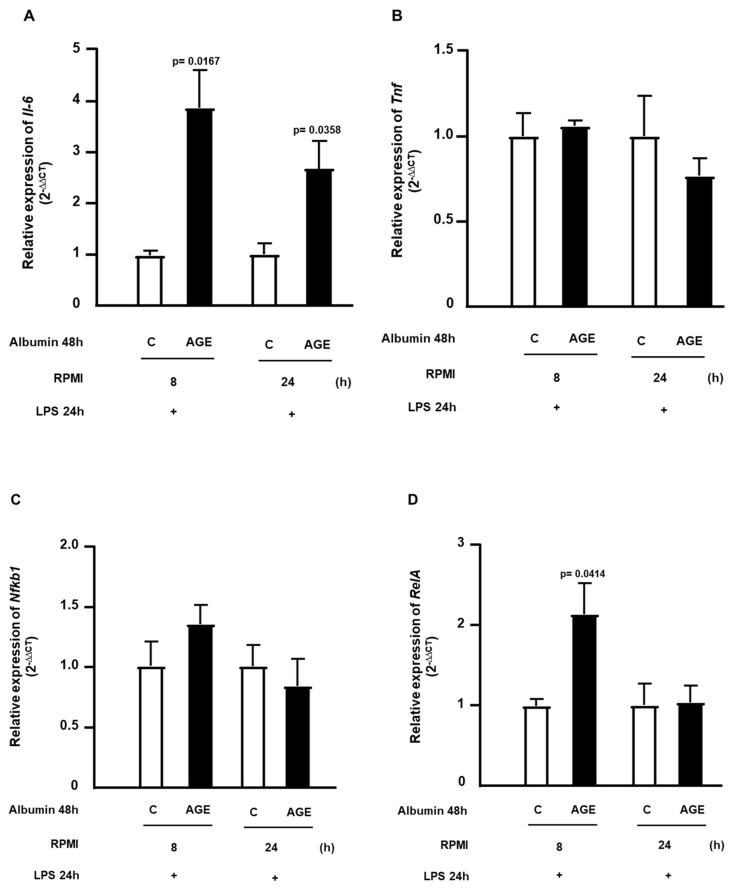
Temporal profile of gene expression related to the inflammatory response in macrophages treated with C- or AGE-albumin and challenged with LPS. RAW264.7 macrophages were treated for 48 h with C- or AGE-albumin (2 mg/mL) and, after washing, maintained over time only in a culture medium containing fatty acid-free albumin. Subsequently, they were challenged with LPS for 24 h. Gene expressions of *Il6* (**A**), *Tnf* (**B**), *Nfkb1* (**C**), and *RelA* (**D**) were assessed by RT-qPCR. Comparisons were made using the Student’s *t*-test, with data normalization to the control condition for each time point; values are presented as mean ± standard error of the mean (*n* = 5).

**Figure 5 ijms-25-02713-f005:**
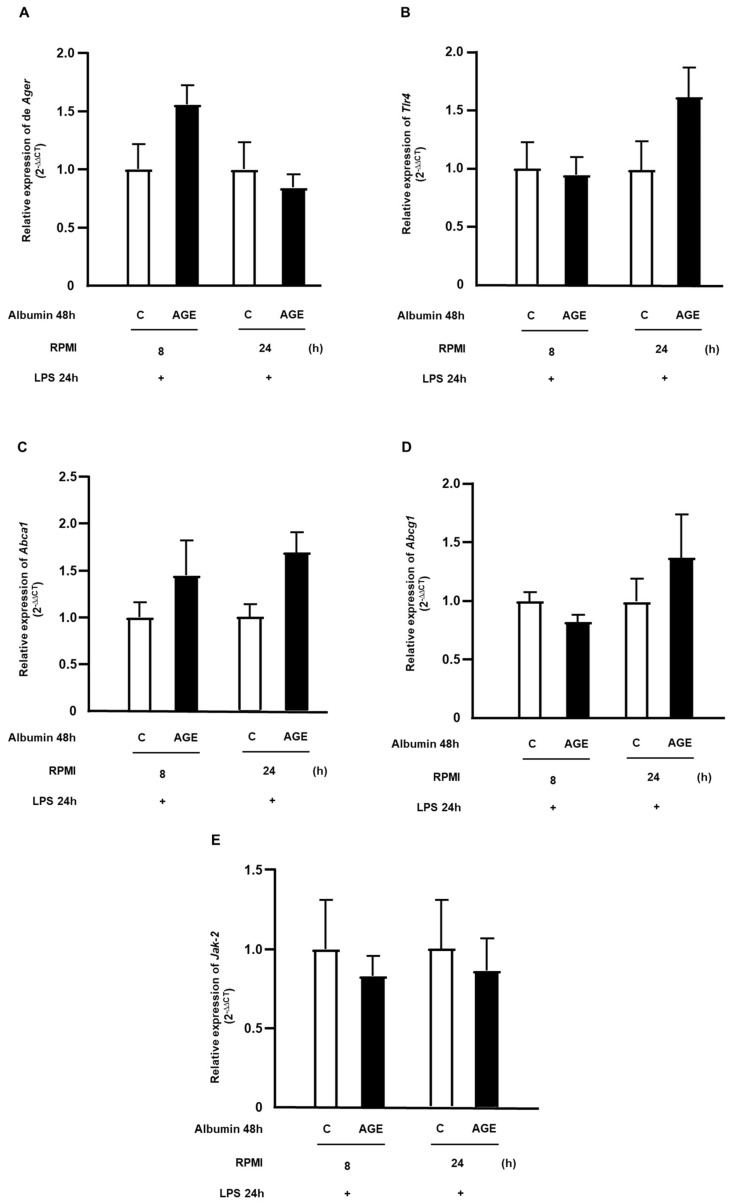
Temporal profile of gene expression related to cholesterol efflux in macrophages treated with C- or AGE-albumin and challenged with LPS. RAW 264.7 macrophages were treated for 48 h with C- or AGE-albumin (2 mg/mL) and, after washing, maintained over time only in a culture medium containing fatty acid-free albumin. Subsequently, they were challenged with LPS for 24 h. Gene expressions of *Ager* (**A**), *Tlr4* (**B**), *Abca1* (**C**), *Abcg1* (**D**), and *Jak2* (**E**) were assessed by RT-qPCR. Comparisons were made using the Student’s *t*-test, with data normalization to the control condition for each time point; values are presented as mean ± standard error of the mean (*n* = 5).

**Figure 6 ijms-25-02713-f006:**
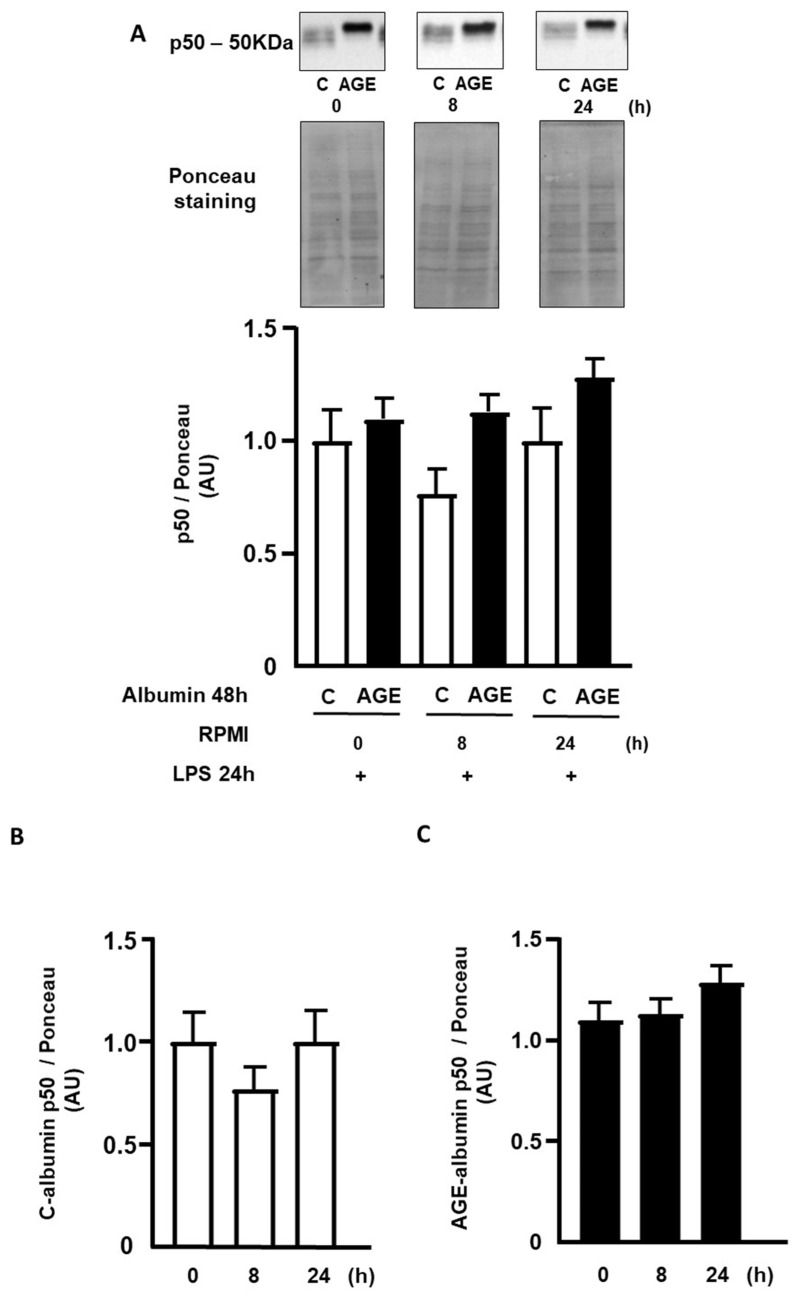
Nuclear contents of p50 and p65 subunits of NFKB in macrophages treated with C- or AGE-albumin and challenged with LPS. RAW 264.7 macrophages were treated with C- or AGE-albumin (2 mg/mL) for 48 h, followed by washing and maintenance over time in a culture medium containing fatty acid-free albumin. Subsequently, they were challenged with LPS (1 µg/mL) for 24 h. The determination of the nuclear contents of p50 and p65 subunits of NFKB was performed by Western blot, and the quantification of band intensity was determined by optical densitometry, normalized to their respective controls (Ponceau staining). Panels (**A**,**D**): p50 and p65 proteins analyzed at each time point (0 h, 8 h, or 24 h) comparing the treatment with C- or AGE-albumin (Shapiro–Wilk normality test, followed by Student’s *t*-test, with values presented as mean ± SEM; *n* = 5). Panels (**B**,**C**,**E**,**F**): comparisons made among different time points (0 h, 8 h, or 24 h) under the same treatment (C or AGE-albumin) using the Shapiro–Wilk normality test, followed by one-way ANOVA (values presented as mean ± standard error of the mean (*n* = 5). AU = arbitrary unit.

## Data Availability

All data reported are included in the manuscript, and raw data can be kindly shared upon personal request to the corresponding author M.P. (m.passarelli@fm.usp.br).

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
