# Peer review of "The Prolonged Activation of the p65 Subunit of the NF-Kappa-B Nuclear Factor Sustains the Persistent Effect of Advanced Glycation End Products on Inflammatory Sensitization in Macrophages"

_ijms, 2024, doi:10.3390/ijms25052713_

Round 1

Reviewer 1 Report

Comments and Suggestions for Authors

The reviewed manuscript is devoted to an interesting and important topic corresponding to the issue of the journal. New interesting results have been obtained. At the same time authors of the paper were in some cases negligent in the design of the manuscript. The paper can be published in the journal after correcting the noted shortcomings listed in the attached file.

Author Response

The reviewed manuscript is devoted to an interesting and important topic corresponding to the issue of the journal, and new interesting results on persistent effect of AGE on macrophage inflammatory priming have been obtained. It is necessary to underline that pharmacological therapies focused on AGE and NFKB inflammatory signaling may help to prevent the effects of AGE mediating inflammatory stress

The authors express their gratitude to the reviewer for their insightful comments and valuable suggestions, which have greatly contributed to enhancing the quality of the manuscript. Please, find below a detailed response addressing each comment. The revisions have been highlighted in the latest version of the manuscript.

At the same time, authors of the paper were in some cases negligent in the design of the manuscript. The paper can be published in the journal after correcting the noted shortcomings, which are listed in the attached file.

The first major note is the following. The authors of the article do not indicate the source of the blood used in the work anywhere. It is even difficult for the reader to understand whether it was human or animal blood, especially since data reported in different articles both on humans and on various animals is discussing. Age and sex is not written as well. All these points must be specified in the “Materials and Methods” section.

The authors have improved the description in the Materials and Methods as follows:

After a 12-hour fasting period, venous blood was drawn from healthy human donors, and plasma was promptly obtained through centrifugation at 3000 rpm, 4°C. All participants signed informed consent, which was previously approved by the Ethics Committee of Hospital das Clinicas (HCFMUSP) da Faculdade de Medicina da Universidade de São Paulo (CAPPesq #2.397.639). Preservatives were added to the plasma pool (µL/mL of plasma): 20 µL of chloramphenicol/gentamicin (0.25%) (Merck, Darmstadt, Germany); 5 µL of benzamidine (2 mM) (Sigma-Aldrich, Steinheim, Germany), 5 µL of aprotinin (0.5%) (Sigma-Aldrich, Steinheim, Germany), and 0.5 µL of PMSF (phenylme-thylsulfonyl fluoride) (Sigma-Aldrich, Steinheim, Germany), followed by density adjust-ment with potassium bromide. Low-density lipoproteins (LDL; d = 1.019 – 1.063 g/mL) and high-density lipoproteins (HDL; D = 1.063 – 1.21 g/mL) were isolated by ultracentrif-ugation in a discontinuous density gradient [22], followed by sterilization through a 0.22 µm filter. Protein concentration was determined using the Lowry´s method [23].

The second major note is connected with the 1st one. If experiments were made with use of human or animal material, the article should include the permissive information from a relevant organization, e.g., from “Bioethical Committee”. This information is indicated in such sections as “Institutional Review Board Statement” at the end of the article.

All participants signed an informed consent, which was previously approved by the Ethics Committee of Hospital das Clinicas (HCFMUSP) da Faculdade de Medicina da Universidade de São Paulo (CAPPesq#2.397.639).

And some minor notes.

Most of the “Abstract” includes a description of the methods used. Only at the end one can read about results of the work. It should have been in reverse. The Abstract have to be mostly specified on the main results of the work.

The abstract has been rewritten in accordance with the reviewer's suggestion (please see below), making it more informative and focused on the study results.

Abstract:

Advanced glycation end products (AGEs) prime macrophages for lipopolysaccharide (LPS)-induced inflammation. We investigated the persistence of cellular AGE-sensitization to LPS, considering the nuclear content of p50 and p65 nuclear factor kappa B (NFKB) subunits and the expression of inflammatory genes. Macrophages treated with either control (C) or AGE-albumin were rested for varying intervals in medium alone before being incubated with LPS. Comparisons were made using one-way ANOVA or Student t-test (n=6). AGE-albumin primed macrophages for increased responsiveness to LPS, resulting in elevated levels of TNF, IL-6, and IL-1 beta (1.5%, 9.4%, and 5.6%, respectively) compared to C-albumin treatment. Secretion of TNF, IL-6, and IL-1beta persisted for up to 24 h even after the removal of AGE-albumin (with an area under the curve greater by 1.6, 16, and 5.2 times, respectively). The expression of Il6 and RelA was higher 8 h after albumin removal, and Il6 and Abca1 was higher 24 h after albumin removal. The nuclear content of p50 remained similar, but p65 showed a sustained increase (2.9 times) for up to 24 h in AGE-albumin-treated cells. The prolonged activation of the p65 subunit of NFKB contributes to the persistent effect of AGEs on macrophage inflammatory priming, which could be targeted for therapies to prevent complications based on the AGE-RAGE-NFKB axis.

The reference to Havel at al. (1955) (Line 133) is absent in the List of References.

The authors have corrected the reference list as was duly noted.

Generally, mentioning cited authors of the methods used in the section “Materials and Methods” was not standardized. In some cases, the author's name and year are used, sometimes – name without year, and sometimes only the number in brackets. It will be better to unify this point.

The authors have corrected the citations as was duly noted.

Some of Figures are too long. The word “long”, but not “large” is specially used. Figures 4 and 5 stretch across three pages, and only on the 3rd page one can read a legend to the figure. It is not very convenient to readers.

The authors agree with the reviewer's comment and have divided the figure to make it more suitable for readers. All other figures have been renumbered.

Finally, it is necessary to edit the List of References according to the rules of the journal: to highlight certain places with bold or italic prints. Please check instructions for authors!

The authors have reviewed the reference list in accordance with the guidelines provided by the journal.

Most of these comments are technical and could be corrected rather easily.

After correcting manuscript according to the notes and comments, it can be published in the journal.

The authors would like to thank the reviewer for the comments and suggestions that help to improve the manuscript.

Reviewer 2 Report

Comments and Suggestions for Authors

The topic is very relevant, since the authors have demonstrated that AGE-albumin primed macrophages for increased responsiveness to LPS, elevating TNF, IL-6, and IL-1 beta  compared to C-albumin. Secretion of TNF, IL-6, and IL-1beta persisted for up to 24h even after the removal of AGE-albumin. The expression of Il6 and RelA (8 h after albumin removal) and Il6 and Abca1 (24h after albumin removal), was higher; the nuclear content of p50 remained similar, but p65 showed a sustained increase for up to 24h in AGE-albumin-treated cells.

The methodology is very modern, using numerous molecular biology biomarkers.

The results have revealed that AGEs sensitize macrophages to inflammatory stimulation promoted by LPS, leading to increased secretion of inflammatory cytokines; the inflammatory sensitization induced by AGE-albumin is prolonged, persisting up to 24 h after removal of AGE from the culture medium; the secretion of IL-6, TNF, and IL-1 beta-induced by LPS after treatment with AGE-albumin indicates the activation of different inflammatory pathways, including the inflammasome system. These results may open new research directions in pharmacological therapies focused on reducing AGE signaling as well as NFKB inflammatory signaling, which may help to prevent the deleterious effects of AGE mediating inflammatory stress.

The conclusions are consistent with the evidence and arguments presented.

The references are very relevant, including also some relevant author’s previous experience in the field.

I suggest some minor editing corrections

1.       Detailed names of different biomolecules should be done at first use

2.       Line 135 (refer) authors must mention the reference nr

3.       References should be written according to Authors Guide for MDPI journals: authors separated by ;, titles of Journals italicized…

Author Response

  1. Detailed names of different biomolecules should be done at first use.

The authors have included the complete names of all biomolecules and proper abbreviattions

  1. Line 135 (refer) authors must mention the reference nr

The authors have included the reference number as properly observed.

  1. References should be written according to Authors Guide for MDPI journals: authors separated by ;, titles of Journals italicized…

The authors have reviewed the reference list in accordance with the guidelines provided by the journal.